# Developmental transcriptomics of the Firebrat: Exploring developmental expression patterns and morphology during the embryogenesis of *Thermobia domestica*

**Wouter P. D. Makkinje** [ID]*, **Esther te Lindert-Blommert** [ID], **Robin van Velzen**, **Eric Schranz** [ID]

Biosystematics group, Wageningen University & Research, Wageningen, The Netherlands.

* wouter.makkinje@wur.nl

## Abstract

Understanding the development of early-diverging lineages is crucial for inferring evolutionary context in evolutionary developmental biology. *Thermobia domestica* (the firebrat), a member of Zygentoma, holds particular significance to insect phylogenetics due to its position as a sister group to all winged insects (Pterygota). We explore its development by reporting on the embryonic morphology using DAPI staining at 14 selected timepoints throughout development, which lasts 10 days. At each timepoint, RNA sequencing was conducted to perform a large-scale transcriptomic analysis. Differential gene expression analysis and gene ontology (GO) enrichment studies revealed global expression patterns and linked biological processes to specific developmental stages. Key findings include the identification of major transcriptional turning points during three developmental stages: The maternal to zygotic transition between 16 and 24 hours after egg-laying (hAEL), katatrepsis between 72 and 96 hAEL, and hatching between 216 and 240 hAEL. Additionally, the GO study mapped the timing of biological processes, such as cleavage, blastoderm formation, dorsal closure, and organogenesis, particularly tracheal and muscular development. These insights establish a robust temporal framework for *T. domestica* embryogenesis and provide a foundation for inferring ancestral and derived developmental traits in insects. The dataset serves as a valuable library for genetic studies and reference for comparative evo-devo studies with other insect species.

## Introduction

Studying the development of early-diverging extant lineages is necessary to provide context for evolutionary developmental (evo-devo) biology. Insects serve as a great model for evo-devo studies as they comprise over half of all described species, display widely varying characteristics and have a long evolutionary history with origins

**Data availability statement:** All relevant data are within the article and its Supporting Information files. Additionally, the Transcriptome Shotgun Assembly project has been deposited in the GenBank database under the accession number GLBB00000000 (https://www.ncbi.nlm.nih.gov/nuccore/GLBB00000000). The version described in this paper is the first version, GLBB01000000.

**Funding:** The author(s) received no specific funding for this work.

**Competing interests:** The authors have declared that no competing interests exist.

dating back to at least 420 million years ago [1,2]. Gene expression is an important driver of embryonic development, regulating cell differentiation and serving as the link between genotype and phenotype [3]. Therefore, revealing developmental gene expression patterns of an early-diverging insect like the ametabolous firebrat, *Thermobia domestica* Packard (Insecta: Zygentoma, Lepismatidae), provides key insight into insect evolution [4].

*T. domestica* is the model organism of choice for early-diverging insect studies due to its phylogenetic position and ease of rearing [5]. As a sister group to all winged insects (Pterygota), Zygentoma serve as a crucial reference point for identifying ancestral insect traits [6,7]. While these early-diverging lineages do not inherently display more ancestral characteristics than derived lineages, as they represent the same evolutionary time since their common ancestor as more derived groups, traits shared across independent early-diverging lineages are likely to be ancestral. Identifying ancestral characteristics establishes expectations for other, more derived species and provides interesting avenues to explore when these expectations are not met. Unsurprisingly, *T. domestica* is a key subject in investigating the origins of wing development and has been key in studying among others but not limited to embryonic evolutionary development, metamorphosis, neurogenesis, sex differentiation and circadian rhythms [8–13]. A testament to their ancestral nature is the absence of metamorphosis, or ametabolism, another critical aspect of insect evo-devo studies [14,15]. These efforts have driven the scientific community to establish several molecular tools, including in situ hybridisation (ISH), RNA interference (RNAi), CRISPR-Cas9, reference genes for qPCR and a sequenced genome [4,5,9,16–18].

Previous studies describing large-scale insect developmental transcriptomics have detected turning points in gene expression. One of those shifts is attributed to the maternal-to-zygotic transition (MZT) [19,20]. During early embryogenesis, the embryo transitions from a dependency on maternally deposited mRNA to zygotic mRNA and lasts until blastoderm cellularisation in *Drosophila melanogaster* [21–23]. One of the genes responsible for these dynamics is *smaug* (*smg*), which is involved in maternal transcript destruction [24]. A second turning point occurs around an insect developmental phenomenon called katatrepsis, where the embryo changes its anteroposterior axis in relationship to the egg by making a 180° turn [25].

With a large-scale developmental transcriptomic analysis on *T. domestica* embryogenesis, we examine developmental dynamics and generate data that can serve as a basis for comparative studies to resolve what is ancestral and what is derived in insect development. In addition, we address the research question: Can large-scale transcriptomics be utilised to resolve the timing of biological processes? To this end, we performed RNA-seq on 14 embryogenic stages. We translate this to a high-resolution temporal expression atlas and provide morphological descriptions of each sequenced timepoint. A differential gene expression (DGE) analysis and subsequent gene ontology (GO) enrichment study confirm the accuracy of our expression atlas and provide insights to the timing of developmental biological processes.



## Materials and methods

### Rearing and egg collection

*Thermobia domestica* populations were obtained through DutchRana (Heerlen, The Netherlands) and kept in perforated containers at 37 °C. Enclosures were provided with egg cartons, a water reservoir with a mesh lid, TetraMin® fish food flakes and rolled oats. Cotton pads were provided for oviposition and screened for egg collection after an egg-laying period of 9 hours. Thus, all our samples represent pooled samples within a 9-hour time range and sample names are based on the upper end of this range. The first two sampling timepoints were collected directly after the egg-laying period at 9 hAEL and 7h later at 16 hAEL. Sample time-points three to seven were subsequently collected at 12h intervals and sample timepoints eight to fourteen were sampled at 24h intervals. The number of eggs collected per sample varied based on age, as less RNA could be extracted from younger eggs. Three biological replicates were sampled for each time-point and eggs were stored in RNA*later*™ (Invitrogen, Waltham, MA, USA) at –80 °C. A complete sampling overview is provided in S1 Table. Additionally, for each sample collected for transcriptome analysis, a separate sample containing ~10 eggs was collected to visualise the stage of development using DNA-specific fluorescent dye, 4′,6-diamidino-2-phenylindole dihydrochloride (DAPI, Sigma-Aldrich, St. Louis, MO, USA), following the fluorescence nuclear staining protocol.

### Fluorescence nuclear staining and visualisation

Eggs were collected in phosphate-buffered saline/Tween-20 (0.1%; PBT) and heat-treated for 5 minutes at 99 °C. Subsequent steps differed based on age due to egg shell maturation. Eggs younger than 60 hAEL were fixed overnight at 4 °C in 9% paraformaldehyde (PFA) – PBT under constant inversion mixing on a rotary shaker. After three rounds of washing with PBT under constant inversion mixing for 5 min, eggs were manually dechorionated using fine forceps. Dechorionated eggs were serially concentrated in increments of 25% to 100% MeOH and stored at −20 °C until further use. Stored eggs were serially concentrated to 100% PBT, washed three times with Milli-Q® water (MQ; MilliporeSigma, Burlington, MA, USA) and permeabilised for 2 hours at room temperature (RT) using a 20 µg/mL Proteinase K (ProtK) – MQ solution. Eggs aged 60 hAEL and older were fixed overnight in Carnoy's solution (55% ethanol, 27% chloroform, 9% glacial acetic acid, 9% ferric chloride) under constant inversion mixing at 4 °C. After three rounds of washing with PBT under constant inversion mixing for 5 min, eggs were subjected to 6 rounds of sonication for 3 seconds at 10% maximum amplitude using an ultrasonic processor model VC 130 PB (Sonics & Materials, Inc, Newtown, CT, USA).

Subsequently, eggs were manually dechorionated, serially concentrated to 100% MeOH and stored at −20 °C until further use. Upon imaging, stored eggs were serially concentrated to 100% PBT and post-fixed in 9% PFA-PBT overnight at 4 °C. After three rounds of washing in MQ, vitelline membranes were manually removed and eggs were permeabilised for 2 hours at RT using a 20 µg/mL ProtK-MQ solution. Eggs were washed with PBT for three rounds prior to staining with a 350ng/mL DAPI-PBT solution for 20 minutes at RT. Eggs were mounted in 85% glycerol on a CoverWell™ imaging chamber with a depth of 0.6mm (Grace Bio-Labs, Bend, OR, United States). Microscopic images were taken using a Zeiss AxioImager.Z2 microscope with a Zeiss Axiocam 506 mono camera and a 60N-C 1" 1.0x camera adapter (Carl Zeiss AG, Oberkochen, Germany). To visualise DAPI fluorescence, excitation and emission filters were set to 358nm and 461nm, respectively. Image processing and stacking were performed with FIJI and CombineZP (v1.0) using Soft stacking or Pyramid stacking, respectively [26].

### RNA extraction and sequencing

Samples were thawed on ice and RNA*later*™ was removed. A total of 550 µL of TRI Reagent® (ZymoResearch, Irvine, CA, USA) was added in two rounds, each round followed by homogenisation using a pestle. Subsequently, the contents

were triturated using a syringe (25G; BD, Franklin Lakes,NJ, USA) to further homogenise the tissue. After vortexing and letting the contents settle for 5 minutes, 150 μl chloroform was added and mixed by inverting the tube for 20 seconds. The mixture was left to settle for 3 minutes, followed by centrifuging at 11,000 RPM for 15 minutes at 4 °C. The supernatant was then collected and used for step 2 of the RNeasy® Mini Kit (Qiagen, Valencia, CA, USA) protocol, continuing onwards. The lysate mixtures of samples that required pooling were centrifuged on the same column during step 3 of the RNeasy® Mini Kit protocol. cDNA library preparation and short-read (150 bp) paired-end (PE) sequencing using the Illumina Nova-Seq 6000 system were performed by Novogene (Cambridge, UK).

## Transcriptome assembly and functional annotation

FastQC v0.11.9 [27] Quality assessment indicated that Novogene performed sufficient quality filtering and trimming. SortMeRNA v4.3.4 [28] with provided rRNA databases was used to filter out rRNA sequences. Reads needed re-pairing afterwards, which was accomplished using the BBMap, Repair.sh release 9/11/2016 script [29]. Contamination was removed by kraken2 v2.1.2 [30] with the nt Database (5/2/2023), filtering out any hits falling outside Bilateria and "No hits" and subsequently using the Univec database [31] following the default BLAST [32] parameters of the VecScreen protocol to filter out any remaining vector contamination. Reference-based transcriptome assembly yielded very low quality scores and was therefore omitted. Therefore, *de novo* transcriptome assembly was conducted using SPAdes v3.15.5 [33] with default parameters. Post-processing consisted of removing transcripts with an open reading frame encoding less than 66 amino acids using TransDecoder.LongOrfs v5.5.0 [34]. Redundancy was removed by means of evidence-based filtering, using the tr2aacds.pl tool from EvidentialGene v2022.01.20 [35]. Lowly expressed genes were filtered out using edgeR's v3.42.4 [36] filterByExpr tool (min.count = 10, min.total.count = 15).

Quality of the transcriptome was assessed using BUSCO v5.3.0 [37] with the insecta_odb10 database and transcriptome mode and TrinityStats.pl from the Trinity package [38]. Sample correlation of the biological replicates was assessed using Trinity's PtR tool [38], creating a PCA plot and a distance matrix only including the top 5,000 variable genes. Homology-based functional annotation of the transcriptome was performed using BLASTX and BLASTP [32] results against protein databases from FlyBase (dmel-all-translation-r6.48.fasta) [39] and Swissprot [40] and Pfam [41] (databases accessed on 13 Apr 2023) hits using HMMER's v3.3.2 hmmscan [42] on translated CDS regions. Results were imported by Trinotate [43] into an SQLite database. GO [44,45] terms associated with annotation results were manually retrieved from the FlyBase and Swissprot online web interfaces and retrieved automatically by Trinotate for Pfam hits. Each transcript was assigned a single annotation with priority given to FlyBase, Swissprot and Pfam annotations in order.

## Differential expression analysis

Gene abundance estimation and DGE analyses were performed using Trinity's [38] Transcript Quantification and Differential expression analysis pipelines, respectively. Alignment-free abundance estimation was performed using Salmon v1.7.0 [46]. Raw counts, trimmed mean of M (TMM)- and transcripts per million (TPM)-normalised expression matrices were generated using the abundance_estimates_to_matrix.pl tool from Trinity [38]. The voom function from the limma R package v3.46.0 [47] was incorporated to conduct pairwise sample gene expression level comparisons on TMM normalised transcript counts and select significant DEGs based on a fold change of 4 and a false discovery rate (FDR) p-value of <1e-3. The resulting selection of DEGs were clustered hierarchically with a Euclidean distance-based measure and complete-linkage method using the analyze_diff_expr.pl script as part of the Trinity package [38]. Clusters were defined with the define_clusters_by_cutting_tree.pl tool by cutting the resulting dendrogram with a K of 45, which is based on a manual assessment of separation of expression patterns and number of genes per cluster. Gene expression results were log2 normalised and centered prior to visualisation using a heatmap created with the R pheatmap package v1.0.12 [48] or cluster-based line graphs using ggplot2 v3.4.3 [49].

### GO enrichment and systematic analysis

GOseq v1.42.0 [50] was utilised by the _runGOseq.R script as part of Trinity [38] to perform cluster-specific GO enrichment analysis, incorporating all ancestral terms of biological process-related GO terms associated with each transcript. GO terms with an over-represented p-value and false discovery rate (FDR) of both < 0.05 were selected as significant enrichments within each cluster. Retrieved GO terms were summarised using an adapted GOslim script [51] with a GOslim [44,45] database manually curated to include mainly terms related to embryonic development, modified from goslim_drosophila.obo release 2023-01-01.

To analyse GO enrichment results for patterns of interest, we systematically examined each cluster and GOslim relevant to embryonic development. First, we excluded clusters without significantly enriched GOslims. Next, we identified clusters uniquely enriched for specific GOslims or sharing enrichments across two or three clusters and assessed the relationship between their expression patterns and the associated GOslims. We then grouped clusters by GOslim enrichment to evaluate coherence in expression patterns, excluding GOslims with highly variable associations. From the remaining observations, we selected those that were collectively associated and best represented major developmental processes.

## Results

### Temporal developmental stage definition

DAPI staining on whole eggs revealed the stage of development at each of the 14 selected timepoints throughout embryogenesis, lasting 10 days (Fig 1). Stage definitions and the developmental context for each timepoint are described in Table 1. The descriptions draw largely upon the findings of the development of another member of the Zygentoma, *Lepisma saccharina* [52]. Therefore, some aspects of development are described but not discernible in our images.

Stage definitions and descriptions of the development of *T. domestica* at each selected timepoint in hours after egg-laying (hAEL).

### Transcriptome assembly and differential expression analysis

RNA sequencing of 42 samples consisting of 14 developmental timepoints with 3 biological replicates each resulted in a total of 3.58 billion raw reads. These were subjected to quality filtering and *de novo* assembled into a raw transcriptome consisting of 1,220,375 contigs. Several subsequent filtering steps yielded a final transcriptome containing 101,100 isoforms with 57,775 putative genes. Quality analysis revealed that the putative gene set has an N50 of 4,284 and a BUSCO completeness of 91.0% (of which 1,119 complete single-copy, 124 complete duplicated, 52 fragmented and 72 missing BUSCOs). This relatively complete transcriptome was subsequently used for a differential expression (DE) study. A principal component analysis (PCA) on transcript abundance estimations revealed grouping of biological replicates and separation between timepoints (Fig 2A). In addition, three transcriptional turning points were identified: (1) during the MZT, between 16 and 24 hAEL; (2) around katatrepsis, between 72 and 96 hAEL; and (3) in the final stages of development, between 216 and 240 hAEL. These turning points are indicated by arrows (Fig 2A). Differential gene expression (DGE) analysis identified 23,554 differentially expressed genes (DEGs). Hierarchical clustering of these DEGs based on temporal expression pattern similarities resulted in 45 clusters displaying distinct developmental expression patterns, which are shown in the heatmap in Fig 2B and displayed separately in S1–S4 Figs.

### Maternal to zygotic transition

Clusters 37, 40, 42, 43, 44 and 45 show the highest over-representation of transcripts annotated as maternally deposited (>38%) and share highly similar expression patterns throughout the early stages of development. These transcripts were extracted as a subset and their expression is displayed in Fig 3 as a visual representation of the maternal to zygotic



**Fig 1. The embryogenesis of *Thermobia domestica* visualised with DAPI staining. (A-M)** Lateral views of DAPI-stained embryos at successive sampled timepoints in hours after egg-laying (hAEL). **(N)** Brightfield image in colour of a hatched *T. domestica* nymph. Scale bar (bottom right) = 100 μm. ant, antenna; cc, cercus; cf, caudal filament; gd, germ disc; l1-3, prothoracic, mesothoracic, and metathoracic legs, respectively; lb, labium; md, mandible; mx, maxilla; pce, protocephalon; pco, protocorm; ser, serosa. Numbers indicate abdominal segments.

**Table 1. Temporal developmental stage definitions of *Thermobia domestica*.**

| Timepoint | Stage | Description | Figure |
|---|---|---|---|
| 9 hAEL | Early cleavage | The first cleavage nuclei, or energids, have divided mitotically. | Fig 1A |
| 16 hAEL | Late cleavage | Energids migrated to the periplasm and several superficial cleavages have passed. | Fig 1B |
| 24 hAEL | Blastoderm | The periphery of the egg is now mostly populated by nuclei as cellularisation (not shown) forms the blastoderm. | Fig 1C |
| 36 hAEL | Germ disc formation | Blastodermal cells at the posterodorsal pole of the egg coalesce to form the germ disc which slightly invaginates and remaining superficial nuclei differentiate into the serosa. | Fig 1D |
| 48 hAEL | Germ band formation | The germ disc has progressed into a germ band while further contracting dorsally to sink into the yolk as the protocephalon and protocorm become distinguishable. | Fig 1E |
| 60 hAEL | Germ band elongation | The germ band elongates sequentially as segments are added to its posterior end and buds of appendages have formed. | Fig 1F |
| 72 hAEL | Katatrepsis | The process of katatrepsis has commenced as the embryo shifts anteriorly to the ventral side of the egg and appendages have become elongated and articulated. | Fig 1G |
| 96 hAEL | Katatrepsis completion | The embryo reached its final position and has completely emerged from the yolk as katatrepsis is completed. | Fig 1H |
| 120 hAEL | Initiation provisional dorsal closure | The amnion migrates over the periphery of the egg (not shown), replacing the serosa which condenses anterodorsally to form the dorsal organ (not shown). | Fig 1I |
| 144 hAEL | Initiation dorsal closure | The lateral dorsal body walls of the embryo extend and merge, starting at the caudal end and progressing anteriorly, initiating dorsal closure. | Fig 1J |
| 168 hAEL | Provisional dorsal closure | The dorsal organ (not shown) has sunk into the yolk and is completely replaced by the amnion, completing the provisional dorsal closure as definitive dorsal closure proceeds. | Fig 1K |
| 192 hAEL | Definitive dorsal closure | The dorsal body walls have merged over the entire dorsal side of the embryo, enclosing the yolk and completing definitive dorsal closure. | Fig 1L |
| 216 hAEL | Maturation | During the final stages the embryo fully matures and the outer cuticle noticably thickens. | Fig 1M |
| 240 hAEL | Hatching | The fully formed embryo hatches as a nymph. | Fig 1N |

transition (MZT). In addition, the expression levels of *smg* were added, which has an expression pattern closely matching that of the selected transcripts and is grouped within cluster 42. Based on these expression patterns, there is a transition towards depletion of maternally deposited transcripts between 16 and 24 hAEL, coinciding with blastoderm formation.

## Developmental GO enrichment analysis

Functional annotations of transcripts were used to perform a cluster-specific GO enrichment analysis, revealing the processes occurring at different stages of development. We have highlighted key observations from the extensive enrichment results, concentrating on four major developmental processes: cleavage, blastoderm formation, dorsal closure, and organogenesis. To enable the exploration of relevant avenues not covered here, complete information on GO enrichment frequencies is provided in S2 Table.

**Cleavage and blastoderm formation.** Three GO terms linked to cleavage and blastoderm formation, and only enriched in clusters showing expression patterns consistent with the expected stages, include nucleus organisation (GO:0006997), nucleocytoplasmic transport (GO:0006913), and cellularisation (GO:0007349). These clusters and their associated expression patterns and GOslim enrichments are displayed in Fig 4. Transcripts belonging to clusters 42 and 43 see the highest expression in the first stages of development, shifting in expression at around 36 hAEL and are enriched for nucleus organisation (GO:0006997). Up until 36 hAEL, nuclei exist as energids, cleaving and migrating through the cytoplasm before forming the blastoderm, fitting this GO term. In addition, cluster 43, along with clusters 36, 37, and 40, shows enrichment for nucleocytoplasmic transport (GO:0006913). These clusters collectively display an increase in expression mostly between 9 hAEL and 36 hAEL, with cluster 36 displaying a peak in expression at 24 hAEL, during cleavage and as energids cellularise to form the blastoderm. Finally, clusters 43 and 45 are enriched for

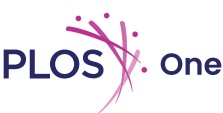

**Fig 2. Differential expression of *Thermobia domestica* development. (A)** Principal components 1 and 2 of a principal component analysis (PCA) plot depicting sample relatedness of biological replicates within sequenced timepoints in hours after egg-laying (hAEL). Arrows depict turning points in gene

expression. **(B)** A heatmap displaying TMM- and log2-normalised, scaled gene expression values, hierarchically clustered based on gene expression pattern similarity. Dendrogram and cluster numbers are shown on the left and right sides of the heatmap, respectively.

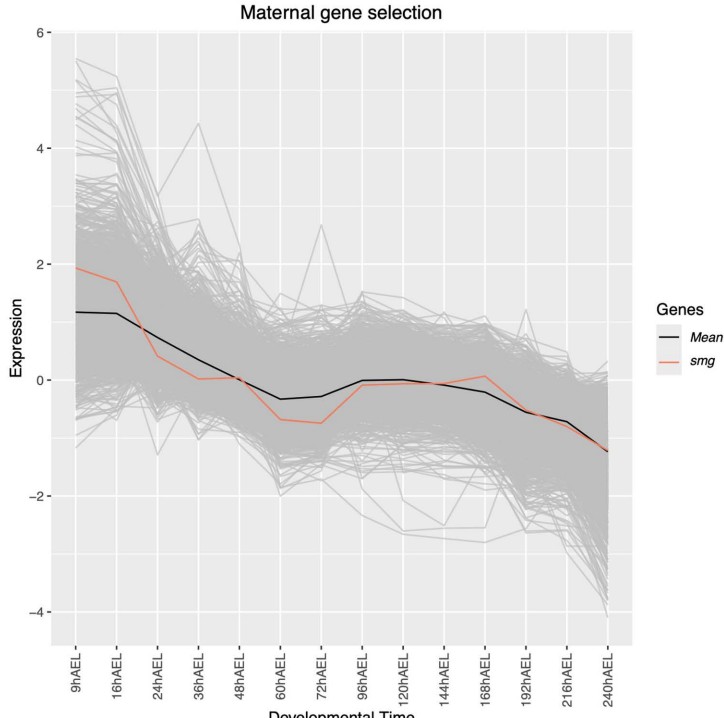

**Fig 3. Expression patterns of transcripts representing the maternal to zygotic transition (MZT).** The expression levels of transcripts from clusters 37, 40, 42, 43, 44, and 45, annotated as maternally deposited throughout development in hours after egg-laying (hAEL). These are selected based on their association with the MZT and the expression of *smaug* (*smg*), associated with maternal transcript depletion, is highlighted. Mean expression values are indicated in black. Expression data is presented in log2-centred TMM (+1) values.

cellularisation (GO:0007349) and highly expressed from cleavage up to blastoderm formation, between 9 hAEL and 24 hAEL.

**Dorsal closure.** Dorsal closure requires cell migration and the joining of extracellular matrices. GOslim enrichments associated with clusters that display notable expression patterns that reflect these dynamics are cell motility (GO:0048870), microtubule-based movement (GO:0007018) and extracellular matrix organisation (GO:0030198). Clusters enriched for cell motility display different expression patterns: early high expression in cluster 45 (Fig 4), late high expression in cluster 19 (Fig 5), or both early and late high expression, as in cluster 33 (Fig 5) and 43 (Fig 4). In clusters 43 and 45 the signal for cell motility is shared with microtubule-based movement. Collectively, these clusters exhibit an overall late high expression starting at 96 hAEL and ending around 192 hAEL, coinciding with the developmental timing of provisional and definitive dorsal closure. The early high expression between 9 and 24 hAEL could be attributed to the migration of energids described in the previous section, as it likely involves similar gene activation. Finally, extracellular matrix organisation is enriched in clusters 19 and 6 (Fig 5) and also display high expression at around 96 and 192 hAEL, with cluster 6 exhibiting a peak in expression in the final stages of dorsal closure at 192 hAEL.

**Organogenesis.** Enriched for the GOslim term embryonic organ development (GO:0048568), cluster 19 broadly reflects the timing of organogenesis at post-katatrepsis between 96 and 192 hAEL (Fig 5). While cluster 43 shares this

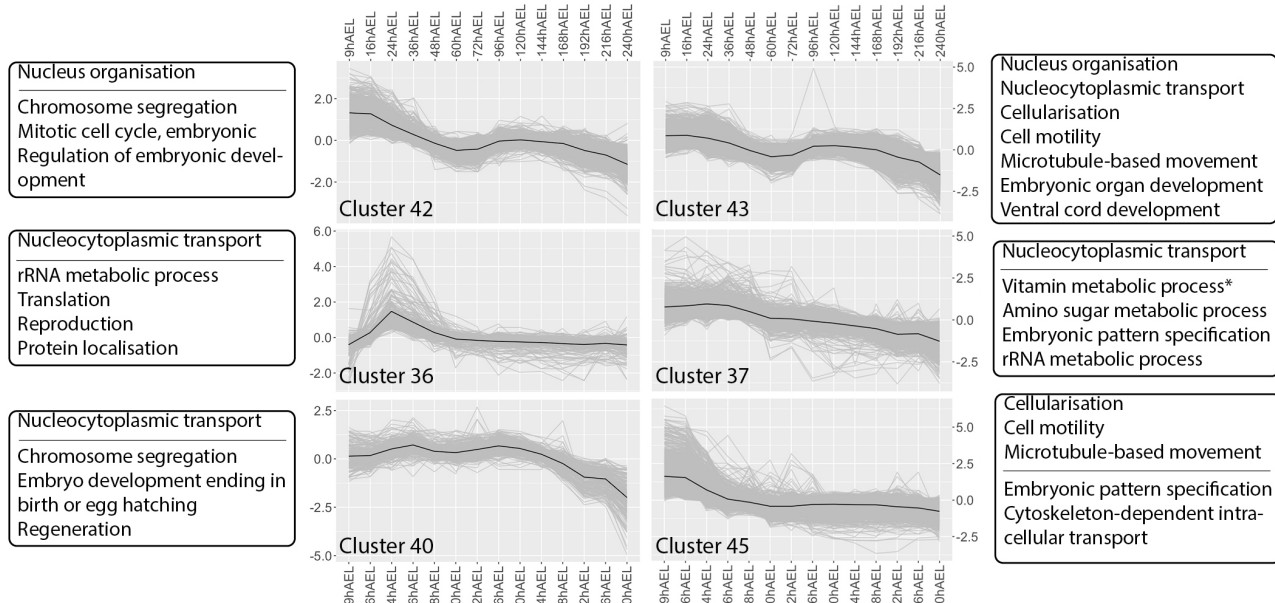

**Fig 4. Expression patterns and GOslim enrichments associated with clusters 36, 37, 40, 42, 43 and 45.** Selected GOslim enrichments are displayed based on relevance to the described processes. Terms above the horizontal line are described in the main text. Stars (*) indicate that the cluster is uniquely enriched for the concerning term. Expression data throughout development in hours after egg-laying (hAEL) is presented in log2-centred TMM (+1) values. Mean expression values are indicated in black.

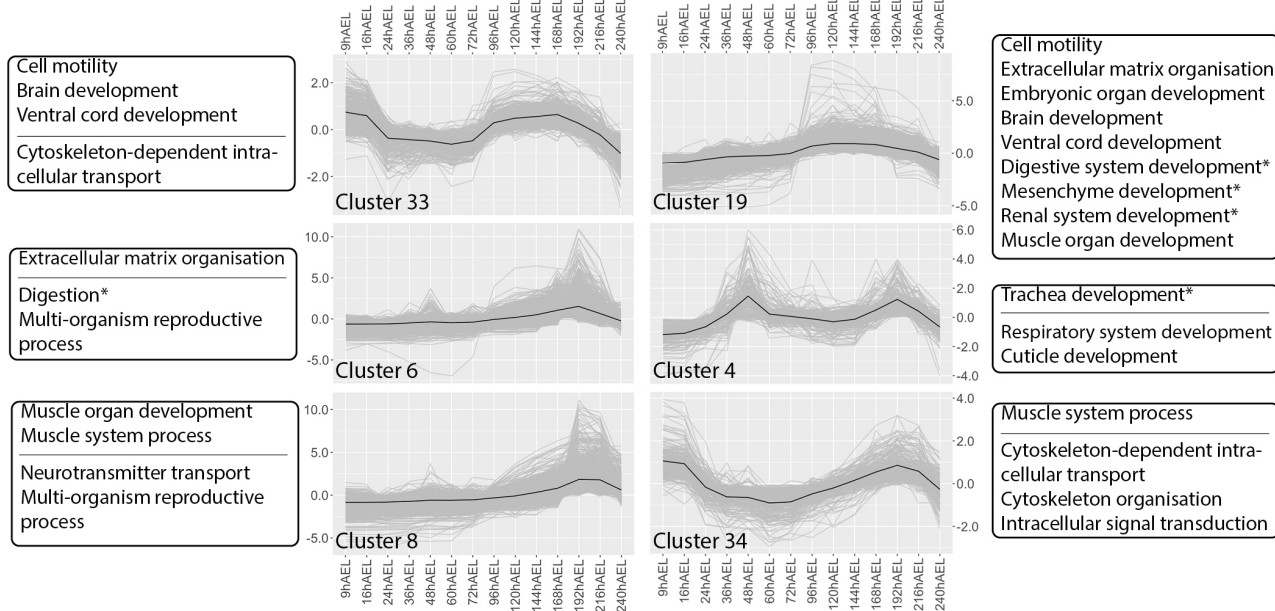

**Fig 5. Expression patterns and GOslim enrichments associated with clusters 4, 6, 8, 19, 33 and 34.** Selected GOslim enrichments are displayed based on relevance to the described processes. Terms above the horizontal line are described in the main text. Stars (*) indicate that the cluster is uniquely enriched for the concerning term. Expression data throughout development in hours after egg-laying (hAEL) is presented in log2-cent-red TMM (+1) values. Mean expression values are indicated in black.

enrichment and displays a slight increase in expression during the same stages, its early activation pattern is likely attributed to early patterning instead of active formation (Fig 4). This is supported by the unique enrichments cluster 19 has for the development of specific organ systems, including digestive system development (GO:0055123) and renal system development (GO:0072001), and that of the tissue involved in organ formation, mesenchyme development (GO:0060485). In addition, clusters 19 and 43 share enrichment for ventral cord development (GO:0007419) with cluster 33 (Fig 5), while clusters 33 and 19 share enrichment for brain development (GO:0007420). An interesting dual peak expression pattern is linked to trachea development (GO:0060438) due to its unique enrichment in cluster 4, with peaks at 48 and 192 hAEL. Lastly, muscle development appears to occur in the later stages, with the first signs of increase in expression at 96 hAEL by cluster 19 and a shift towards high expression around 120 hAEL indicated by clusters 8 and 34, with strongly elevated expression towards the end of development as evidenced by the enrichment of muscle organ development (GO:0007517) and muscle system process (GO:0003012) in these clusters (Fig 5).

## Discussion

### Accuracy of developmental staging

The optimised DAPI staining protocol effectively identified developmental stages of sequenced timepoints, even though the resolution was insufficient to distinguish minute details or cell types. The timing of developmental stages also closely aligns with findings from Truman *et al.*, 2024 and Lv *et al.*, 2024 [15,53].

As mentioned above, descriptions of development draw upon the findings of another member of the Zygentoma, *Lepisma saccharina*, and these species were found to be highly similar in their developmental morphology, stages and progression. Although one exception to this is the extent to which the embryo sinks into the yolk and assumes a concave position during anatrepsis around 48 to 60 hAEL. Within our imaged samples, we did not encounter this. Rather, the embryo seems to remain at the periphery of the egg, similar to another Zygentoma member, *Ctenolepisma longicaudata*, which is why we refrained from using the term anatrepsis in our overview [13]. However, we can't rule out the possibility that this phenomenon was missed or insufficiently stained and therefore excluded from our results. Other descriptions of the embryogenesis of *T. domestica* also do not report on the extent of anatrepsis [15,54].

### Assessment of the differential gene expression analysis

A considerable number of genes, 23,554, have been identified as significantly differentially expressed. This is likely due to a large number of putative genes present in the transcriptome caused by over-assembly, characteristic of *de novo* transcriptome assembly. In addition, the high resolution of our sampling setup increases the number of measurement instances, thereby enhancing the likelihood of detecting differential expression of a gene.

Despite requiring a 9-hour margin to collect sufficient eggs for RNA extraction, the PCA demonstrates the quality of our sampling setup and subsequent sequencing (Fig 2A). It also reveals some global expression patterns throughout development. Three turning points in gene expression are illustrated: during the blastoderm formation (16-24 hAEL), during katatrepsis (60-72 hAEL), and during the final maturation stages before hatching (216-240 hAEL). Fig 3 shows that most transcripts annotated as maternally deposited in *Drosophila melanogaster* start depleting between 16 and 24 hAEL, which is substantiated by the expression of *smg*. Similar observations were made in the Ephemeropteran *Ephemera vulgata*, where the maternal-to-zygotic transition (MZT) was established at 24 hAEL based on similar analyses [19]. The first transcriptional turning point therefore likely represents the MZT. The second transition was previously described in similar large-scale developmental transcriptomic studies on an Odonata species, *Ischnura elegans* and an Ephemeroptera species, *E. vulgata* [19,25]. These studies report on a strong transcriptional shift around katatrepsis, highlighting the significance of this stage in early-diverging insect embryogenesis. Our findings align with this phenomenon, indicating that it is shared across early-diverging insects, and suggesting that this shift is an ancestral trait within insects. The final transition is straightforward, as the insect progresses from the embryonic to the nymphal stage.

## GO enrichments reveal the timing of biological processes

Our GO enrichment analysis associated specific biological processes with expected developmental timepoints as defined by our morphological overview. This alignment between transcriptomic data and morphology validates the accuracy of our expression atlas. To avoid ambiguous results from our GO enrichment analysis, we focused on terms that were either unique to one cluster or shared between a few clusters with similar expression patterns. We only report on these and a selection of other relevant GO terms in the main text, while an overview of all enrichment frequencies is presented in S2 Table. We acknowledge that this selection may introduce bias and encourage other experts in the field to leverage our dataset to reveal other patterns of interest. GOslims were used to condense the numerous enrichments into broader terms, at the cost of specificity, as genes involved in different parts of biological processes are grouped under parent terms. Although this may appear to be a limitation, the extensive data generated by our comprehensive analysis and the numerous processes occurring during development must be condensed in some manner to allow for the interpretation of global developmental expression patterns.

While many enrichments for biological processes exhibited clear associated expression patterns, the GO enrichment analysis has limitations. Some enrichments were shared across multiple clusters with varying expression patterns. For example, the GOslim for cuticle production is enriched in eight clusters (4, 6, 8, 13, 15, 24, 37, and 39; S2 Table). *T. domestica* is known to deposit two cuticles during embryogenesis, one around 72 hAEL and another between 204–216 hAEL [54]. The enriched clusters collectively represent two distinct increases in expression: one between 9–72 hAEL and another between 192–216 hAEL. While this loosely aligns with our expectations, we consider the results inconclusive due to the conflicting expression patterns associated with the GOslim enrichment.

Cellularisation was enriched in clusters 43 and 45, with high expression during the first stages of development (Fig 4). This pattern corresponds to cellularisation during blastoderm formation, as confirmed by morphological observations [55]. However, a second wave of expression was exhibited by cluster 45, aligning with the timing of organogenesis at 96 hAEL as shown by cluster 19 (Fig 5). This observation can be explained by the cellularisation of energids present in the yolk of the primary midgut, which occurs during midgut epithelium formation in *T. domestica* [56]. This demonstrates how certain processes are involved in different developmental events, resulting in non-specific temporal expression signals.

Another example is found in the observations on cell motility (GO:0048870). This GO term encompasses many mechanisms cells employ to induce movement and is therefore involved in, among others, dorsal closure, energid migration and neurogenesis [55,57–59]. The latter becomes apparent in our data as clusters 33 and 19 share enrichment for brain development (GO:0007420) and cell motility (GO:0048870; Fig 5). Although we hypothesised an increase in expression levels of genes involved in cell motility during katatrepsis, this was not observed. The exact biological processes responsible for the movements observed during katatrepsis, besides the involvement of the extra-embryonic membranes, remain to be identified [60]. While inconclusive, our observations indicate that the mechanisms underlying the cellular movements necessary for dorsal closure are distinct from those driving katatrepsis.

According to Rost *et al.*, 2005, organogenesis starts on the fourth day of embryogenesis, aligning with our observations of a shift towards high expression of associated clusters between 72 and 96 hAEL [56]. The expression pattern associated with trachea development (GO:0060438) displayed by cluster 4 is highly similar to what was found in the GO enrichment analysis on the developmental transcriptome of *Ephemera vulgata* (Fig 5) [19]. In both cases, an isolated peak in expression is found at 48 hAEL, which coincides with the germ band stage in both species, albeit slightly earlier in *T. domestica*, followed by an increase in expression during later stages in development. Additionally, the onset of muscle development aligns in these species, occurring after provisional dorsal closure at 120 hAEL in *T. domestica* and 132 hAEL in *E. vulgata* [19]. Similar patterns were observed in the development of Odonata, where early formation is followed by a rapid progress of muscle development after dorsal closure [25,61]. These alignments in transcriptional observations between species on muscle and trachea development suggest that they are ancestral to insect development.

   



## Conclusion

This study provides a comprehensive analysis of developmental expression patterns and morphology during the embryogenesis of *T. domestica*. By integrating morphological observations with transcriptomic data, our study links biological processes to specific developmental stages and their timepoints. These biological processes include cleavage, blastoderm formation, dorsal closure, and organogenesis, particularly tracheal and muscular development. Major transcriptional turning points were identified during three developmental stages: The MZT, katatrepsis, and hatching. The alignment of these observations, particularly regarding organogenesis and transcriptional turning points, with similar studies on more derived insect species suggests that these features are ancestral to insect development. These findings establish a robust temporal framework for understanding embryogenesis in the evolutionary significant insect that is *T. domestica*. Therefore, this dataset serves as a vital reference for comparative studies with other insect species, such as *Ephemera vulgata* and *Ischnura elegans*, to further explore aspects of insect evo-devo and determine which developmental characteristics are ancestral to insects.

## Supporting information

**S1 Table. *Thermobia domestica* egg sampling overview.** Information on the dates and number of eggs sampled for each sample of the temporal developmental expression atlas study. Sample names ending with numbers were pooled. ELP, egg-laying period; hAEL, hours after egg-laying.
(PDF)

**S1 Fig. Expression levels of transcripts belonging to clusters 1–12.** Expression data throughout development in hours after egg-laying (hAEL) is presented in log2-centered TMM (+1) values. Mean expression values are indicated in black.
(PDF)

**S2 Fig. Expression levels of transcripts belonging to clusters 13–24.** Expression data throughout development in hours after egg-laying (hAEL) is presented in log2-centered TMM (+1) values. Mean expression values are indicated in black.
(PDF)

**S3 Fig. Expression levels of transcripts belonging to clusters 25–36.** Expression data throughout development in hours after egg-laying (hAEL) is presented in log2-centered TMM (+1) values. Mean expression values are indicated in black.
(PDF)

**S4 Fig. Expression levels of transcripts belonging to clusters 37–45.** Expression data throughout development in hours after egg-laying (hAEL) is presented in log2-centered TMM (+1) values. Mean expression values are indicated in black.
(PDF)

**S2 Table. Frequencies of enriched GOslims for each cluster. Rows list all enriched GOslims in alphabetical order, while columns represent clusters.** Clusters 2, 3, 5, 7, 9, 10, 11, 18, 20, 25–27, 29, 30 and 32 were not significantly enriched for any GO term and were therefore excluded from the overview.
(XLSX)

## Acknowledgments

The authors would like to thank the members of the Department of Entomology for providing a site for firebrat rearing; the members of the Laboratory of Biosystematics and Genetics for their aid in lab work; Teun de Jong for testing nuclear staining methods; and Sivasubramani Selvanayagam for the discussions and assistance in the bioinformatics analyses.



## Author contributions

**Conceptualization:** Wouter P. D. Makkinje.

**Data curation:** Wouter P. D. Makkinje, Esther te Lindert-Blommert.

**Formal analysis:** Wouter P. D. Makkinje.

**Funding acquisition:** M. Eric Schranz.

**Investigation:** Wouter P. D. Makkinje, Esther te Lindert-Blommert.

**Methodology:** Wouter P. D. Makkinje, Esther te Lindert-Blommert.

**Project administration:** Robin van Velzen, M. Eric Schranz.

**Supervision:** Wouter P. D. Makkinje, Robin van Velzen, M. Eric Schranz.

**Validation:** Wouter P. D. Makkinje, Robin van Velzen.

**Visualization:** Wouter P. D. Makkinje, Esther te Lindert-Blommert.

**Writing – original draft:** Wouter P. D. Makkinje.

**Writing – review & editing:** Robin van Velzen, M. Eric Schranz.

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
