## [Decision Letter · Decision Letter 0]

7 Feb 2025

PONE-D-25-00977Developmental transcriptomics of the Firebrat: Exploring developmental expression patterns and morphology during the embryogenesis of *Thermobia domestica*PLOS ONE

Dear Dr. Makkinje,

Thank you for submitting your manuscript to PLOS ONE. After careful consideration, we feel that it has merit but does not fully meet PLOS ONE’s publication criteria as it currently stands. Therefore, we invite you to submit a revised version of the manuscript that comprehensively addresses the points raised during the review process.

We look forward to receiving your revised manuscript.

Kind regards,

Michael Schubert

Academic Editor

PLOS ONE

Journal Requirements:

Reviewers' comments:

Reviewer's Responses to Questions

**Comments to the Author**

1. Is the manuscript technically sound, and do the data support the conclusions?

Reviewer #1: Yes

Reviewer #2: Partly

2. Has the statistical analysis been performed appropriately and rigorously? 

Reviewer #1: No

Reviewer #2: I Don't Know

3. Have the authors made all data underlying the findings in their manuscript fully available?

Reviewer #1: Yes

Reviewer #2: No

4. Is the manuscript presented in an intelligible fashion and written in standard English?

Reviewer #1: Yes

Reviewer #2: Yes

5. Review Comments to the Author

Reviewer #1: Summary:

The study investigates the developmental transcriptomics of Thermobia domestica (the firebrat), an early-diverging insect species. The researchers used RNA sequencing at 14 time points across the 10-day embryogenesis period to examine gene expression patterns. Key findings include three major transcriptional turning points: (1) the maternal-to-zygotic transition (MZT) between 16-24 hours after egg laying (hAEL), (2) katatrepsis between 72-96 hAEL, and (3) hatching between 216-240 hAEL. Gene ontology (GO) enrichment studies linked specific biological processes such as cleavage, blastoderm formation, dorsal closure, and organogenesis to these developmental stages. The data establishes a robust temporal framework for gene expression during T. domestica embryogenesis, which can aid in understanding ancestral and derived traits in insect development.

I previously reviewed a very similar manuscript from the same group, working on a different species. My criticism remains the same, although I will state some aspects more strongly now, since this is the second paper of the same type, and there may be more coming in the future. My criticism does not necessarily reflect on the decision whether to publish the manuscript or not. The manuscript is OK as it stands, especially for publication in PLoS One. I urge the authors to make whatever changes they can, with the understanding that some changes are not practical, and to think differently about this type of data collection, should they have additional such projects in the pipeline.

First, this is a data paper. The data presented are important and useful and the data collection was done to a high level. However, the analyses the authors carry out do not provide much added value, and should be presented as a “pilot study” to assess the quality of the data. Essentially, the authors cluster the genes in the transcriptome based on similar expression profiles. They then pick GO terms that fit their expectations for the processes that should be taking place at the equivalent stages. There is no real null hypothesis and no statistical comparison of the GO terms they chose relative to a random selection of GO terms. In addition, GO term analysis provides information on the putative biological function of the genes in a given cluster. However, this information must be handled carefully, given the possibility of research biases in the GO terms themsleves, such as above-average community interest in certain biological functions. This can lead to an above average of available comparative information on such a topic (and associated genes). This can lead to incorrect cluster characterization. Whole-mount in situ hybridization data (as a future project) that provide spatial expression data within developing embryos are thus required to verify (or falsify) GO analysis predictions.

Second, the quality of the images is well below what is expected from an embryological paper. I understand that these images are provided only as reference, and this is not meant to be a descriptive developmental paper, but some of the images are uninterpretable (especially Figures 1C,E,J, but the others are not much better). In my review of this group’s previous paper I encouraged the authors to find access to a confocal microscope. In that manuscript, embryos were hard to come by. However, in this case, the authors have access to a breeding colony and could easily collect additional embryos and make an effort to get better images.

Third, if I understood correctly, all of the transcriptomic analyses are not of individual time points, but of pooled samples within a relatively long time range. This is not stated explicitly, and has significant implications for the interpretation of the data. The fact that there are ranges and not discrete time points also explains why the authors see relatively smooth transitions between the samples in the PCA analysis. This is because there is bound to be some overlap between samples due to inter and intra clutch variability in development.

Specific comments:

L.43-45 - A date of origin isn’t in and of itself a reason to study a group

L.45-47 - Gene expression is only one of several important development drivers.

L.47-50 - The study of basally-branching groups doesn’t provide a foundation. It could provide context or hint at an original role, mechanism, or network structure. The foundation for insect evo-devo studies has been laid for the past 30 or so years.

L.51-53 - The reasoning is upside down, “due to its phylogenetic position and ease of rearing, T. domestica has long been a model for… and a suite of experimental tools including… has been developed.”

L.67 - “... in expression…”→“... in gene expression…”

Material and Methods - please clarify protocols and use specific language.

Heat-treated →boiled

L.104- how were the eggs agitated? Rocking? Shaking?

Throughout the manuscript - abbreviated scientific names should be used only when the full name is given relatively recently. I suggest giving the full scientific name once per major section (Introduction, Methods. etc.) When E. vulgata and I. elegans reappeared in the final paragraph, I had to scroll all the way back to the beginning to remember what the genus name was.

Reviewer #2: Determining the pattern of transcription during embryogenesis of Thermobia provides very valuable information on an insect that is of great phylogenetic interest. I appreciate the DAPI-stained images of the various embryonic time-points that the authors have provided. These are a very useful contribution to the paper.

I cannot judge the appropriateness of the methods employed by the authors. I will leave this issue to other reviewers. I found it hard to understand what conclusions come out of the study. The data presented towards the end of the Results appear to be a random selection of possible correlations. It might be more useful to the reader to systematically go through the major changes in expression through development. Perhaps one might start by focusing on clusters that have a single peak of expression (e.g. clusters like 6, 7, 8, 9, 13, 15, 24, 26, etc. ). Arrange the clusters along a developmental timeline and then (based on GO values) relate the types of genes that are expressed at that time to ongoing development. There are also some intriguing clusters that are biphasic (for example: clusters 4, 5, 28, 29, 30, and 38). Being insects, Thermobia deposit two cuticles during embryogenesis (see Konopova and Zrzavy, 2005); one around katatrepsis and the other after definitive dorsal closure. Do any of these clusters include genes that would be involved in cuticle production (e.g., chitin synthetase) or any of the cellular processes involved in cuticle production?

Where possible the authors should compare their data with that presented in the recent paper by Lv YN, Zeng M, Yan ZY, et al., BMC Biol 2024, 22:232, https://doi. org/10.1186/s12915-024-02029-2. This paper also presents data on changing transcript profiles during embryogenesis of Thermobia, although its special focus is on the development of the leg.

Minor issues:

Why are clusters 2 to 11 missing from the supplementary table of GOslim frequencies? They should be included.

Figure 1: Part G: it would be useful to label antenna; H: shift the ant label to side facing the reader; M the embryo is now covered by the second embryonic cuticle. He axis in the lower right makes no sense for a curved embryo – it would be better to omit it.

Figs 3 to 5 and S1

I think that the figures can be improved. Gray on gray makes it difficult to follow the lines. I would suggest black on gray and then using a red line for the average.

Developmental biologists usually expect time along the X-axis to be in constant increments. Although the sample points are uniformly spread the time between them is not a constant: the data points on the first day are about 8 hr apart then the next two days are 12 hr apart followed by 24 hr intervals thereafter. This presents a very distorted view of Development. If not too much trouble, it would be more useful to have real time along the X-axis.

Table 1: 144hAEL. “merge from the caudal end to the apical end”. Since the caudal end is the apical end, I do not know what this means? Perhaps …. merge starting at the caudal end and progressing anteriorly. ???

6. PLOS authors have the option to publish the peer review history of their article (what does this mean? ). If published, this will include your full peer review and any attached files.

**Do you want your identity to be public for this peer review?** For information about this choice, including consent withdrawal, please see our Privacy Policy .

Reviewer #1: **Yes: ** Ariel Chipman

Reviewer #2: No

---

## [Author Response · Author response to Decision Letter 1]

10 Mar 2025

* Response to Academic Editor

The Academic Editor, Dr. Michael Schubert, noted that our manuscript did not fully adhere to PLOS ONE’s style requirements. We have made the necessary revisions and hope the updated formatting meets PLOS ONE’s standards, allowing the editing process to proceed smoothly.

Comment 1: “Please ensure that your manuscript meets PLOS ONE's style requirements, including those for file naming. The PLOS ONE style templates can be found at

https://journals.plos.org/plosone/s/file?id=ba62/PLOSOne_formatting_sample_title_authors_affiliations.pdf”

Reply 1: We have edited the manuscript according to these guidelines.

Comment 2: “Please include captions for your Supporting Information files at the end of your manuscript, and update any in-text citations to match accordingly. Please see our Supporting Information guidelines for more information: http://journals.plos.org/plosone/s/supporting-information.”

Reply 2: Captions for supporting information were included at the end of the manuscript and in-text citations were updated.

* Response to Reviewer 1, Prof. Dr. Ariel Chipman

- Review comments

Comment 1: “I previously reviewed a very similar manuscript from the same group, working on a different species. My criticism remains the same, although I will state some aspects more strongly now, since this is the second paper of the same type, and there may be more coming in the future. My criticism does not necessarily reflect on the decision whether to publish the manuscript or not. The manuscript is OK as it stands, especially for publication in PLoS One. I urge the authors to make whatever changes they can, with the understanding that some changes are not practical, and to think differently about this type of data collection, should they have additional such projects in the pipeline.”

Reply 1: We sincerely appreciate the time and effort the reviewer invested in reviewing both of our manuscripts and are grateful for their recognition that our study meets the publication standards of PLOS ONE and the encouragements to contribute more meaningfully to evolutionary developmental research and strengthen our contribution to the field. We have incorporated as many of the suggestions as practically possible to address their concerns, enhance clarity for our readers, and better articulate the limitations of our study.

Comment 2: “First, this is a data paper. The data presented are important and useful and the data collection was done to a high level. However, the analyses the authors carry out do not provide much added value, and should be presented as a “pilot study” to assess the quality of the data.”

Reply 2: We agree with the reviewer that this is predominantly a data paper and appreciate his acknowledgement of the importance and usefulness of the data presented. Data quality is important, and we devote much attention to showing this in our manuscript. Nevertheless, in our opinion, our study goes beyond the level of a “pilot study” as we relate expression patterns to various biologically relevant aspects of development. Specifically, we reveal patterns resolving the timing of processes such as cleavage, blastoderm formation, dorsal closure, and organogenesis.

While we may have differing perspectives on the added value of identifying which transcriptional developmental trends are ancestral to insects, we recognise that our manuscript may not have conveyed the value it adds specifically, clearly enough. In assessing ancestral developmental processes, many of our findings—both regarding transcriptomic trends throughout embryogenesis and GO enrichments at specific stages—are most meaningful in the context of our previous study on mayfly developmental transcriptomics, published in BMC Genomics, and similar studies on the developmental transcriptomics of other insects, as this gives us an indication of which observations might be ancestral [1,2]. We have reported on several such shared observations to assess ancestrality, throughout the discussion (lines 378-380, 436-438). We have also added a statement in the introduction to emphasise the added value of this information (lines 58-60): “Identifying ancestral characteristics establishes expectations for other, more derived species and provides interesting avenues to explore when these expectations are not met”.

Comment 3: “Essentially, the authors cluster the genes in the transcriptome based on similar expression profiles. They then pick GO terms that fit their expectations for the processes that should be taking place at the equivalent stages. There is no real null hypothesis and no statistical comparison of the GO terms they chose relative to a random selection of GO terms.”

Reply 3: GO enrichment analysis is a statistical procedure to identify GO terms that are significantly overrepresented in a specific cluster, compared to a null hypothesis where GO terms are selected randomly based on their frequency in the data. It therefore is a statistical comparison similar to what the reviewer suggests. Nevertheless, the reviewer correctly points out that GO enrichment analyses are generally descriptive in nature and, as can be seen in our Table in S1 Table, large numbers of enriched GO terms for our transcriptome-scale clusters. Therefore, our approach involved systematically examining all expression pattern/GO associations and selecting those that could offer insights into the relationship between morphological stage determinations and transcriptomic-based determinations. To clarify our selection process and demonstrate that it has more merit than a random selection, we have included a section in the Methods (lines 199–206).

Comment 4: “In addition, GO term analysis provides information on the putative biological function of the genes in a given cluster. However, this information must be handled carefully, given the possibility of research biases in the GO terms themselves, such as above-average community interest in certain biological functions. This can lead to an above average of available comparative information on such a topic (and associated genes). This can lead to incorrect cluster characterization.”

Reply 4: We acknowledge and agree that our selection of enriched GO terms may introduce bias and remains primarily descriptive. To allow other researchers in the field to make their own selections, we provide a comprehensive list of all enriched GO terms in the Table in S1 Table. We have also added a sentence about this in the discussion (lines 389-391).

Comment 5: “Whole-mount in situ hybridization data (as a future project) that provide spatial expression data within developing embryos are thus required to verify (or falsify) GO analysis predictions.”

Reply 5: We agree with the reviewer that in situ hybridisation to assess spatial expression would be an exciting next step and the data we present here can be used to design probes for this purpose. However, as already acknowledged by the reviewer, that is beyond the scope of our current study.

Comment 6: “Second, the quality of the images is well below what is expected from an embryological paper. I understand that these images are provided only as reference, and this is not meant to be a descriptive developmental paper, but some of the images are uninterpretable (especially Figures 1C,E,J, but the others are not much better). In my review of this group’s previous paper I encouraged the authors to find access to a confocal microscope. In that manuscript, embryos were hard to come by. However, in this case, the authors have access to a breeding colony and could easily collect additional embryos and make an effort to get better images.”

Reply 6: We appreciate the reviewer’s encouragement to use confocal microscopy to generate even better images. Indeed, for a descriptive developmental paper, we would have included confocal microscopy images and likely even SEM images. However, such equipment was not directly available to us and the images are primarily provided as a general reference for the sampled developmental stages. We would also like to quote reviewer 2 who explicitly appreciates our images: “I appreciate the DAPI-stained images of the various embryonic time-points that the authors have provided. These are a very useful contribution to the paper.”

Comment 7: “Third, if I understood correctly, all of the transcriptomic analyses are not of individual time points, but of pooled samples within a relatively long time range. This is not stated explicitly, and has significant implications for the interpretation of the data. The fact that there are ranges and not discrete time points also explains why the authors see relatively smooth transitions between the samples in the PCA analysis. This is because there is bound to be some overlap between samples due to inter and intra clutch variability in development.”

Reply 7: We agree with the reviewer on the importance of emphasizing that our samples represent pooled samples within a 9-hour time range, not individual timepoints. This was already explicitly stated in the methods (lines 92 and 93) and in the discussion on the PCA results (lines 364 and 365): “Despite requiring a 9-hour margin to collect sufficient eggs for RNA extraction, the PCA demonstrates the quality of our sampling setup and subsequent sequencing (Fig. 2A)”. To even further clarify this aspect for the reader, we have revised our methods section to include the following statement (lines 93 and 94): "Thus, all our samples represent pooled samples within a 9-hour time range and sample names are based on the upper end of this range ".

- Specific comments

Comment 1: “L.43-45 - A date of origin isn’t in and of itself a reason to study a group.”

Reply 1: We agree that our original statement did not accurately convey our intention in mentioning the date of origin. We have revised it to: “…and have a long evolutionary history with origins dating back to at least 420 million years ago” (lines 46 and 47).

Comment 2: “L.45-47 - Gene expression is only one of several important development drivers.”

Reply 2: We agree and have rephrased our statement (line 47): “Gene expression is an important driver of embryonic development”.

Comment 3: “L.47-50 - The study of basally-branching groups doesn’t provide a foundation. It could provide context or hint at an original role, mechanism, or network structure. The foundation for insect evo-devo studies has been laid for the past 30 or so years.”

Reply 3: The word “foundation” was indeed used incorrectly here. We have revised it to: “…provides key insight into insect evolution” (line 51).

Comment 4: ‘L.51-53 - The reasoning is upside down, “due to its phylogenetic position and ease of rearing, T. domestica has long been a model for… and a suite of experimental tools including… has been developed.”’

Reply 4: We agree with the reviewer that the previous phrasing of this section presented upside down reasoning. To improve on this and the flow of the introduction, we have made the necessary changes (lines 52-55).

Comment 5: ‘L.67 - “... in expression…”→“... in gene expression…”’

Reply 5: We incorporated this change (line 69).

Comment 6: “Material and Methods - please clarify protocols and use specific language.”

Reply 6: We have incorporated all suggested clarifications.

Comment 7: “Heat-treated →boiled”

Reply 7: While we appreciate the attentiveness, we prefer using the term “heat-treated”, as the contents of the tube were not brought to a boiling point at the onset of the 5-minute exposure to 99°C.

Comment 8: “L.104- how were the eggs agitated? Rocking? Shaking?”

Reply 8: We have clarified the means of agitation to “inversion mixing on a rotary shaker” (line 109) and “inversion mixing” (lines 110, 116 and 117).

Comment 9: “Throughout the manuscript - abbreviated scientific names should be used only when the full name is given relatively recently. I suggest giving the full scientific name once per major section (Introduction, Methods. etc.) When E. vulgata and I. elegans reappeared in the final paragraph, I had to scroll all the way back to the beginning to remember what the genus name was.”

Reply 9: We have adjusted the abbreviations for Ephemera vulgata (lines 376, 429 and 452), Ischnura elegans (lines 452), Drosophila melanogaster (lines 72 and 370), Ctenolepisma longicaudata (line 353) and Lepisma saccharina (line 348).

* Response to Reviewer 2

- Review comments

Comment 1:”I appreciate the DAPI-stained images of the various embryonic time-points that the authors have provided. These are a very useful contribution to the paper.”

Reply 1: We sincerely thank the reviewer for their appreciation of the included DAPI-stained images.

Comment 2: “I found it hard to understand what conclusions come out of the study. The data presented towards the end of the Results appear to be a random selection of possible correlations. It might be more useful to the reader to systematically go through the major changes in expression through development. Perhaps one might start by focusing on clusters that have a single peak of expression (e.g. clusters like 6, 7, 8, 9, 13, 15, 24, 26, etc. ). Arrange the clusters along a developmental timeline and then (based on GO values) relate the types of genes that are expressed at that time to ongoing development. There are also some intriguing clusters that are biphasic (for example: clusters 4, 5, 28, 29, 30, and 38).”

Reply 2: We appreciate and understand the concern that the selection of correlations may appear random. We made an effort to condense the large amount of information into an easy-to-follow results section of observations that link morphology to transcriptomic data. In analyzing GOslims, we followed a selection process very similar to what the reviewer suggests. We recognize that our lack of explanation regarding the methods used to identify these correlations has caused confusion, as reflected in similar comments from Reviewer 1 (Comment 3). To address this, we have expanded the Methods section on GO enrichment analysis (lines 199–206).

Regarding the mentioned biphasic clusters, we’ve discussed the noteworthy GO enrichments for cluster 4 (lines 333-335); however, the other clusters had no associations we could confidently link to specific biological processes or no associations at all, as shown in the table below.

Cluster Enriched GO terms

5 NA

28 generation of precursor metabolites and energy, mitochondrion organisation, protein-containing complex assembly

29 NA

30 NA

38 lipid metabolic process, cellular metabolic process

Comment 3: “Being insects, Thermobia deposit two cuticles during embryogenesis (see Konopova and Zrzavy, 2005); one around katatrepsis and the other after definitive dorsal closure. Do any of these clusters include genes that would be involved in cuticle production (e.g., chitin synthetase) or any of the cellular processes involved in cuticle production?”

Reply 3: We thank the reviewer for their valuable suggestions based on their expertise. Cuticle production is indeed one of the GOslims we found enrichments for, although results on this GOslim were deemed inconclusive. Collectively, the increases in expression levels shared by the associated clusters (between 9-72 hAEL and 204-216 hAEL) resemble the timepoints of cuticle deposition that were reported by Konopova and Zrzavy (2005; between 72 hAEL and 204-216 hAEL)[3]. However, the enrichment is shared by 8 different clusters, displaying distinct expression patterns, presenting contradictory evidence. As part of addressing limitations of our GO enrichment analysis, we included a section in the discussion addressing these findings (lines 397-405).

Comment 4: “Where possible the authors should compare their data with that presented in the recent paper by Lv YN, Zeng M, Yan ZY, et al., BMC Biol 2024, 22:232, (https://doi/.org/10.1186/s12915-024-02029-2. This paper also presents data on changing transcript profiles during embryogenesis of Thermobia, although its special focus is on the development of the leg.”

Reply 4: We thank the reviewer for bringing this interesting

---

## [Decision Letter · Decision Letter 1]

2 May 2025

Developmental transcriptomics of the Firebrat: Exploring developmental expression patterns and morphology during the embryogenesis of *Thermobia domestica*

PONE-D-25-00977R1

Dear Dr. Makkinje,

We’re pleased to inform you that your manuscript has been judged scientifically suitable for publication and will be formally accepted for publication once it meets all outstanding technical requirements.

Kind regards,

Michael Schubert

Academic Editor

PLOS ONE

Reviewers' comments:

Reviewer's Responses to Questions

**Comments to the Author**

1. If the authors have adequately addressed your comments raised in a previous round of review and you feel that this manuscript is now acceptable for publication, you may indicate that here to bypass the “Comments to the Author” section, enter your conflict of interest statement in the “Confidential to Editor” section, and submit your "Accept" recommendation.

Reviewer #1: All comments have been addressed

Reviewer #2: All comments have been addressed

2. Is the manuscript technically sound, and do the data support the conclusions?

Reviewer #1: Yes

Reviewer #2: Yes

3. Has the statistical analysis been performed appropriately and rigorously? 

Reviewer #1: Yes

Reviewer #2: Yes

4. Have the authors made all data underlying the findings in their manuscript fully available?

Reviewer #1: Yes

Reviewer #2: Yes

5. Is the manuscript presented in an intelligible fashion and written in standard English?

Reviewer #1: Yes

Reviewer #2: Yes

6. Review Comments to the Author

Reviewer #1: (No Response)

Reviewer #2: (No Response)

7. PLOS authors have the option to publish the peer review history of their article (what does this mean? ). If published, this will include your full peer review and any attached files.

**Do you want your identity to be public for this peer review?** For information about this choice, including consent withdrawal, please see our Privacy Policy .

Reviewer #1: **Yes: ** Ariel Chipman

Reviewer #2: No

---

## [Editor Report · Acceptance letter]

PONE-D-25-00977R1

PLOS ONE

Dear Dr. Makkinje,

I'm pleased to inform you that your manuscript has been deemed suitable for publication in PLOS ONE. Congratulations! Your manuscript is now being handed over to our production team.

Kind regards,

on behalf of

Dr. Michael Schubert

Academic Editor

PLOS ONE